# Directed Synthesis of Humic and Fulvic Derivatives with Enhanced Antioxidant Properties

Alexander B. Volikov [1], Nikita V. Mareev [1], Andrey I. Konstantinov [1], Alexandra A. Molodykh [1], Sofia V. Melnikova [1], Alina E. Bazhanova [2], Mikhail E. Gasanov [3], Evgeny N. Nikolaev [3], Alexander Ya. Zherebker [3], Dmitry S. Volkov [1,4], Maria V. Zykova [5] and Irina V. Perminova [1,*]

[1] Department of Chemistry, Lomonosov Moscow State University, Leninskie Gory 1-3, 119991 Moscow, Russia; ab.volikov@gmail.com (A.B.V.); mr.n.mareev@yandex.ru (N.V.M.); konstant@org.chem.msu.ru (A.I.K.); alex-molod@yandex.ru (A.A.M.); sonyamelnikova20@gmail.com (S.V.M.); dmsvolkov@gmail.com (D.S.V.)

[2] Department of Soil Science, Lomonosov Moscow State University, Leninskiye Gory 1-12, 119991 Moscow, Russia; iiaalina69@gmail.com

[3] Skolkovo Institute of Science and Technology, Bolshoy Boulevard 30-1, 121205 Moscow, Russia; gasanov.mickhail@gmail.com (M.E.G.); E.Nikolaev@skoltech.ru (E.N.N.); a.zherebker@skoltech.ru (A.Y.Z.)

[4] Department of Chemistry and Physical Chemistry of Soils, V.V. Dokuchaev Soil Science Institute, Pyzhevsky Per., 7/2, 119017 Moscow, Russia

[5] Department of Chemistry, Siberian State Medical University, 634050 Tomsk, Russia; huminolog@mail.ru

* Correspondence: iperm@org.chem.msu.ru

**Abstract:** Redox moieties, which are present in the molecular backbone of humic substances (HS), govern their antioxidant properties. We hypothesized that a directed modification of the humic backbone via incorporation of redox moieties with known redox properties might provide an efficient tool for tuning up antioxidant properties of HS. In this work, hydroquinonoid and hydronaphthoquinonoid centres were used, which possess very different redox characteristics. They were incorporated into the structure of coal (leonardite) humic acids CHA) and peat fulvic acids (PFA). For this goal, an oxidative copolymerization of phenols was used. The latter was induced via oxidation of hydroquinones and hydroxynapjtaquinones with a use of Fenton's reagent. The structure of the obtained products was characterized using NMR and FTIR spectroscopy. H/D labelling coupled to FT ICR mass spectrometry analysis was applied for identification of the reaction products as a tool for surmising on reaction mechanism. It was shown that covalent -C-C- bond were formed between the incorporated redox centers and aromatic core of HS. The parent humic acids and their naphthoquinonoid derivatives have demonstrated high accepting capacity. At the same time, fulvic acids and their hydroquinonoid derivatives have possessed both high donor and high antioxidant capacities. The kinetic studies have demonstrated that both humic acids and their derivatives showed much slower kinetics of antioxidant reactions as compared to fulvic acids. The obtained results show, firstly, substantial difference in redox and antioxidant properties of the humic and fulvic acids, and, secondly, they can serve as an experimental evidence that directed chemical modification of humic substances can be used to tune and control antioxidant properties of natural HS.

**Keywords:** humic acids; fulvic acids; modification; hydroquinones; naphthoquinones; Fenton reagent; antioxidant activity; redox capacity

## 1. Introduction

The antioxidant properties of natural humic substances (HS) attract substantial attention due to their importance for both the biological activity of HS and the mediating effects in microbial and photochemical reactions [1–4]. In the benchmark publication by Aeschbacher et al. [4], the authors applied electrochemical approach for the direct measurement of both the donor- and accepting capacities of HS [4]. The systematic electrochemical measurements undertaken on standard samples of the International Humic Substances Society (IHSS) isolated from leonardite, soil, peat, and freshwater, enabled assessment of the

natural variation range of donor and acceptor capacities of HS: the highest donor capacity was observed for freshwater HS, the lowest one—for the leonardite HA [5,6]. At the same time, the leonardite HA were characterized with the highest acceptor capacity [5,6]. The obtained data were important not only for understanding the natural variations in donor and accepting capacity of HS. They enabled structure—redox properties and mechanistic studies on natural HS. As a result, photo-oxidation was related to the changes in electrochemical properties of HS [7], the molecular basis of natural polyphenolic antioxidants was proposed [8], biogeochemical redox transformations of natural organic matter (NOM) and HS as well as iron cycling were explained [9–13] and substantial progress was achieved in understanding contaminants' biotransformation [14,15]. The dominant role of aromatic structural units, nominally, titratable phenols, was unambiguously demonstrated [7–9], providing solid experimental evidence for the long-stated hypothesis on quinonoid moieties as carriers of redox activity of HS [16]. The obtained structure-property relationships are of particular value for mechanistic understanding of redox-behavior of HS in the environment. They enabled predictions on the fate of redox-sensitive contaminants (e.g., Hg(II), Cr(VI), Pu(V, VI), diazo dyes, and others) in the organic-rich environments [7,17–19].

Given the important role of biocatalytic cycles in the redox transformations of contaminants in the environment, the information on redox mediating capacity of HS is of indispensable value [14,17]. Methodical electrochemical approaches for the assessment of mediating properties of HS were developed in another set of publications by Aeschbacher et al. [5,20], who have demonstrated that HS could successfully function as an extracellular electron shuttle enhancing the accessibility of insoluble substrates for microbial redox transformations.

In our previous work [21], we used phenol formaldehyde condensation for incorporation of quinonoid centers into HS backbone aimed at controlling the redox properties of humic materials. The major drawback of this approach is a use of toxic formaldehyde, which prevents its broad application for agricultural and environmental applications. This study is devoted to development of an alternative "green" synthesis of the quinonoid-enriched derivatives, which uses Fenton's reagent for in situ oxidation of hydroquinones to quinones under mild conditions followed by oxidative polymerization of the produced quinones with humic aromatic backbone. The particular attention is given to the studies on antioxidant properties of the obtained derivatives including their kinetic properties. The synthesized derivatives are tested for inhibition of methane synthesis by anaerobic bacterial communities.

## 2. Experimental

### 2.1. Materials and Reagents

Leonardite potassium humate (CHP) and sodium fulvate (PFA) isolated from dissolved organic matter of peat were used as the parent humic materials. Both materials were provided by Humintech GmbH (Grevenbroich, Germany). The following set of compounds was used for modification of the humic matrix: 1,4-hydroxyquinone, 1,2-hydroxyquinone, 1,4-naphthoquinone, 2-hydroxy-1,4-naphthoquinone (all 97% or higher, Sigma-Aldrich, Chemie GmbH, Taufkirchen, Germany), 2-methyl-1,4-hydroxyquinone (97%; ChemMed, Moscow, Russia). Fenton's reagent was prepared in situ using $FeSO_4 \cdot 7H_2O$ and $H_2O_2$ (30%); redox capacity was measured using $K_3Fe(CN)_6$ (all of a reagent grade; ChemMed, Moscow, Russia). Antioxidant activity by TEAC method was measured using Trolox (97%; Sigma-Aldrich, Chemie GmbH, Taufkirchen, Germany) and ABTS (98%; Sigma-Aldrich, Chemie GmbH, Taufkirchen, Germany), and $K_2S_2O_8$ (analytical grade) from ChemMed (Moscow, Russia). Bond Elut PPL SPE cartridges (Agilent, Santa Clara, CA, USA, 5 g, 60 mL) were used for purification. High-purity distilled water (18.2 MΩ) was prepared using a Simplicity 185 system (Millipore, Merck, Darmstadt, Germany). Determination of dissolved organic carbon was carried out using a TOC-L carbon analyzer (Shimadzu, Kyoto, Japan). Potassium biphthalate was used for organic carbon (OC) calibration in

the range of concentrations of 0–20 mg/L. The HS sample was dissolved in water at the concentration of 5–15 mg/L.

*2.2. Modification Protocols of the Humic Materials Used in This Study*

Prior to modification, the insoluble mineral part of CHP parent material was separated by centrifugation. It was not required for completely soluble FA material. A weighed amount of CHP (1 g) was dissolved in distilled water (30 mL) and centrifuged for 5 min at 10,000 rpm yielding ~5% wt of insoluble part. The supernatant was transferred into a 100 mL glass beaker and 0.25 g of hydroquinone (HQ) or naphthoquinone (NQ) were added under continuous stirring under atmospheric air. The pH value of the resulting solution was adjusted to 10 using 40% KOH solution. Then, the obtained solution was added with 30% $H_2O_2$ (2 mL, 0.02 mol) followed by dropwise addition of 10 mL of 1 mM $FeSO_4$. During the iron sulfate addition the solution pH value was kept in the range of 9 to 10 by adding KOH solution as needed. The resulting reaction mixture was transferred into a 100 mL flask equipped with a reflux condenser, and heated for 4 h at 70 °C in a water bath under continuous stirring. The reaction mixture was then cooled down and rotor evaporated at 50 °C to dryness. The similar syntheses were carried with the FA parent material. The obtained fulvic acid derivatives were purified using Bond Elut SPE PPL cartridges as described elsewhere [22,23]. In brief, the SPE PPL cartridge was activated according to producer instructions by passing three volumes of methanol. The FA sample (120 mL, 40 g/L) was acidified to pH 2 and passed through a cartridge. It was eluted with 60 mL of methanol and rotor-evaporated to dryness. A list of the obtained materials is summarized in Table 1.

**Table 1.** Polyphenolic derivatives of humic and fulvic acids obtained in this study.

| Name | Synthesis |
|------|-----------|
| CHP-HQ | coal HA (CHP) and 1,4-hydroquinone |
| CHP-MHQ | coal HA (CHP) and 2-methy-1,4-hydroquinone |
| CHP-PC | coal HA (CHP) and 1,2-hydroquinone |
| CHP-1,4-NQ | coal HA (CHP) and 1,4-naphthoquinone |
| CHP-2-OH-1,4-NQ | coal HA (CHP) and 2-hydroxy-1,4-naphthoquinone |
| FA-HQ | peat FA (FA) and 1,4-hydroquinone |
| FA-MHQ | peat FA (FA) and 2-methyl-1,4-hydroquinone |
| FA-PC | peat FA (FA) and 1,2-hydroquinone |
| FA-1,4-NQ | peat FA (FA) and 1,4-naphthoquinone |
| FA-2-OH-1,4-NQ | peat FA (FA) and 2-hydroxy-1,4-naphthoquinone |

*2.3. Synthesis of D-Labeled Derivatives*

Deuterium labeling was used for studying the modification mechanism. For this purpose, a similar experiment with 1,4-hydroquinone (0.125 g) was performed in $D_2O$ (15 mL) and then a weight of FA (0.5 g) was added under continuous stirring in air. The pH of the obtaining solution was adjusted to 10 using a 40% KOH solution. Then, 1 mL (0.02 mol) of 30% $H_2O_2$ was added under continuous stirring, followed by dropwise addition of 1.5 mL of 1 mM $FeSO_4$. A value of pH was kept in the range of 9 to 10 by adding KOH solution. For control, similar syntheses were performed with FA and 1,4-hydroquinone alone. The first control was synthesized: using FA as the only reagent (without addition of hydroquinone) and was designated as [$^2$H]FA, and the second control was synthesized using hydroquinone (HQ) as the only reagent and was designated as [$^2$H]HQ.

*2.4. Structural Characterization of the Humic Materials Used in This Study*

$^{13}$C-NMR spectra were recorded on an Avance 400 NMR spectrometer (Bruker, BioSpin, Rheinstetten, Germany) located at the Department of Chemistry, Lomonosov MSU, operating at the frequency of 100 MHz using the Carr-Parsell-Maybum-Gill pulse sequence (CPMG). The spectrum sweep width was 42,735 Hz, the registration time of free induc-

tion decay was 0.2 s, the time delay between pulse sequences (Td) was 7.8 s, spectrum acquisition time was 12 h. The samples were prepared by dissolving a weight of 70 mg in 0.6 mL of 0.3 M $NaOD/D_2O$ placed into a 5 mm NMR tube. Fourier transformation was performed with preliminary exponential weighting of a FID-signal with a time constant equivalent to line broadening of 100 Hz.

The IR spectra of the obtained preparations were recorded on a Vertex 70 IR Fourier spectrometer (Bruker Optik GmbH, Ettlingen, Germany) equipped with a GladiATR attachment of broken total internal reflection (ATR) with a diamond crystal (Pike Technologies, Madison, WI, USA). Spectral recording range: 4000–400 $cm^{-1}$, resolution 2 $cm^{-1}$, number of scans of the sample and background—64. To remove atmospheric moisture, the spectrometer was continuously blown with ultra-dry air with a dew point of $-70$ °C. Before recording the IR spectrum, the sample was placed on an ATR crystal, clamped with a screw, and then the spectrum was recorded. The data were processed using OPUS 7.5 software (Bruker Optik GmbH, Ettlingen, Germany).

Optical properties were characterized using UV-Vis and fluorescence spectroscopy. Absorption spectra were recorded on a Cary-50 Probe spectrophotometer (Varian, Palo Alto, CA, USA) operating in the UV-Vis regions (200–800 nm) equipped with a quartz cuvette (optical length of 1 cm). A CLARIO-Star (BMG Labtech Ortenberg, Germany) plate reader was used for kinetic experiments. The measurements were carried out in transparent 96-cell plates. The fluorescence spectra were recorded using using a FluoroLog FL-3-222 spectrofluorimeter (HORIBA, Longjumeau, France). The spectra were recorded at two excitation wavelengths of 280 and 350 nm. The measurements were run in a quartz cell (1 cm). pH values of the solutions were measured using a 713 pH Meter (Metrohm, Herisau, Switzerland) equipped with a universal glass electrode.

### 2.5. FT ICR MS Analysis

D-labeled FA derivatives and control samples were characterized using a FT MS Bruker Apex Ultra mass spectrometer equipped with a harmonized cell (Bruker Daltonics, Bremen, Germany), 7 T superconducting magnet, and electrospray ion source (ESI) operated in negative ionization mode. The FT ICR MS data were processed using the self-made Python scripts. The CHONS formulae were assigned using the following chemical constraints: O/C ratio $\leq 1$, $0.25 <$ H/C ratio $\leq 2$, element counts [C $\leq 120$, H $\leq 200$, $0 < O \leq 60$, N $\leq 2$, S $\leq 1$]; and mass accuracy window < 0.5 ppm. Since the deuteration process was used, two hydrogen isotopes were taken into account in the calculations: $^1H$ and $^2H$. The assigned CHNOS formulae were plotted into van Krevelen diagram [24,25], which represents relationship of the H/C ratio versus the O/C ratio. Open source Matplotlib library (Python) was used for data visualization [26]. The mass lists were juxtaposed for determination of modification of parent ions with fragments of molecular formulae $C_6O_2(^1H + {}^2H)_4$ $(^2H > 0)$, which corresponds to addition of hydroquinone moieties accompanied by the loss of two hydrogen atoms.

### 2.6. Determination of Redox Capacity

The redox capacity of the CHP and FA derivatives was determined according to the reported procedure [27,28] also described in our previous work [21]. The samples of humic derivatives were prepared in 0.07 M phosphate buffer at pH 6 at a concentration of 100 mg/L. A solution of $K_3Fe(CN)_6$ (0.5 mM) was used for determination. The obtained buffer and working solutions in 20 mL tubes were used to prepare solutions A, B, and C. Solution A contained 50 mg/L of the derivative and 0.25 M $K_3Fe(CN)_6$; blank solution B contained 0.25 M $K_3Fe(CN)_6$, and blank solution C contained 50 mg/L of HS derivative. The solutions were stirred and left in the dark for 24 h at room temperature. Then, the optical density of all solutions was measured at a wavelength of 420 nm, which corresponds to the maximum absorption of hexacyanoferrate (III) [28]. A decrease in optical density $\Delta A$ due to the reduction of $K_3Fe(CN)_6$ was calculated by the following formula:

$$\Delta A = A(B) + A(C) - A(A) \tag{1}$$

where A (A), A (B), A (C) are optical densities of the solutions A, B and C, respectively.

The amount of reduced hexacyanoferrate (III) was calculated from a calibration curve constructed using $K_3Fe(CN)_6$ solutions of various concentrations. Redox capacity was calculated using Equation (2):

$$\text{Redox capacity (mmol/g)} = \frac{\Delta\nu(K_3Fe(CN)_6)}{C_{HA}} \times 1000, \qquad (2)$$

where $\Delta\nu(K_3Fe(CN)_6)$ is the amount of recovered $K_3Fe(CN)_6$, calculated according to the calibration curve, mmol/L; $C_{HA}$ is the concentration of HS (initial humic and fulvic acids and their derivatives) in solution, mg/L.

### 2.7. Determination of Antioxidant Capacity of the Humic Materials Using TEAC Method

The antioxidant capacity of the CHP and FA derivatives was determined by the reported TEAC method [29–31]. A working solution of ABTS radical was prepared by dissolving a weight of 11 mg of ABTS in 1 mL of distilled water placed in a 2 mL microtube. The obtained solution was mixed by vortexing for 2 min. Then, 100 μL of 2.45 mM $K_2S_2O_8$ was added to the resulting solution and mixed. After 12 h, the ABTS radical solution was diluted with a phosphate buffer (0.1 M, pH = 7.4) until its absorbance value reached 0.7 at a wavelength of 734 nm. A stock solution of Trolox ((±)-6-hydroxy-2,5,7,8-tetramethylchroman-2-carboxylic acid) was prepared by dissolving a weight of 2.5 mg of Trolox in 500 μL of ethanol placed in a 2 mL Eppendorf microtube. The latter was placed onto Vortex until Trolox was completely dissolved. The working solution of Trolox at a concentration of 100 μM was quantitatively transferred into a 100 mL flask and diluted with water. The solutions of HA and FA derivatives used in this study were prepared at concentration of 100 mg/L by dissolution in a 100 mL volumetric flask. For measurements, calibration solutions were prepared using the Trolox standard [30,31].

For measurements, different aliquots of working solutions of samples were added to the prepared microtubes: 80, 140, and 200 μL and the volume was brought up to 200 μL with distilled water. In a blank experiment, 200 μL of distilled water was used. 1800 μL of ABTS working solution was added to all prepared tubes. After 40 min, absorption spectra were recorded in the range 400–800 nm. Measurements for samples and blank were performed in triplicate. To calculate the antioxidant capacity for each spectrum, the optical absorption at 734 nm was measured, the difference between the absorption of the ABTS radical solution in distilled water (control experiment) and the sample was determined, and the antioxidant capacity was found from the calibration dependence in Trolox equivalents for all samples in units of mmol/g [30–32].

### 2.8. Kinetic Measurements on the Rate of ABTS Radical Quenching by Humic Derivatives

Kinetics of antioxidant properties of HS was studied by different authors [33–35]. In this study, we used procedure described by Klein et al., 2018 [33] based on ABTS radical quenching. A working solution of ABTS radical was prepared by dissolution of a weight of 11 mg of ABTS in 1 mL of distilled water followed by addition of 100 μL of 2.45 mM $K_2S_2O_8$. After exposure for 12 h, the ABTS radical solution was diluted with a phosphate buffer (0.1 M, pH = 7.4) until its absorbance reading reached 0.7 at a wavelength of 734 nm. The solutions of HA and FA derivatives were prepared at concentrations of 60, 80, and 100 mg/L. In a 96-cell plate, an aliquot (20 μL) of working solutions of humic derivatives was added in four replicas; distilled water was used as a control. Then, 180 μL of a working solution of ABTS radical was added to each cell using an eight-channel pipette, and a plate was placed in a UV-Vis reader [33,34]. Absorption was recorded at 734 nm during 40 min with 1 min interval. For each point, a decrease in the content of ABTS radical was determined according to the Equation (3):

$$\Delta ABTS^{\bullet+} = C(ABTS_0^{\bullet+}) \times \frac{A_{kt} - A_{st}}{A_{k_0}} \qquad (3)$$

where $C(ABTS_0{}^+)$ is the concentration of ABTS radical in the working solution, $A_{st}$ is the optical absorption of the sample at a time t, $A_{kt}$ is the optical absorption of the control at a time t, $A_{ko}$ is the optical absorption of control at the starting point of the measurement.

For quantitative assessment of contribution of slow and fast centers into the rate of ABTS quenching by HS derivatives over the exposure time, we used the model developed by Klein et al. [33], who represented reaction rate as a sum of the fast and slow stages of the reaction:

$$\Delta(ABTS^{\bullet+}) = \nu(HS_{fast}) \times (1 - e^{-k_{fast} \times C(ABTS^{\bullet+})^0 \times t}) + \nu(HS_{slow}) \times (1 - e^{-k_{slow} \times C(ABTS^{\bullet+})^0 \times t}) \quad (4)$$

where $\Delta(ABTS^{\bullet+})$ is a change in the ABTS-radical concentration, $\nu(HS_{fast})$ is the portion of fast centers, $\nu(HS_{slow})$ is the portion of slow centers, $k_{fast}$ is the second-order constant of the fast reaction, $k_{slow}$ is the second-order constant of the slow reaction, $C(ABTS^{\bullet+})_0$ is the initial concentration of $ABTS^{\bullet+}$ (at the time = 0), t is the reaction time.

## 3. Results and Discussion

### 3.1. Synthesis and Structural Characteristics of the Humic Derivatives Obtained in This Study

Modification of HS was carried out using oxidative polymerization of phenols. Fenton's reagent was used to generate phenoxyl radicals from the parent phenols as shown in Figure 1a–c for the example of hydroquinone:

**Figure 1.** Schematic reaction pathways for synthesis of quinonoid-enriched humic materials using Fenton's reagent and hydroquinonic and naphthoquinonic modifiers used in this study: (**a**) generation of hydroxyl radical; (**b**) assumed mechanism of interaction between the hydroxyl radical and the phenolic fragment; (**c**) binding of phenolic fragments to the humic aromatic core forming humic copolymer with pendant hydroquinone units; (**d**) three hydroquinones (1,4-hydroquinone, 2-methyl-1,4-hydroquinone, 1,2-hydroquinone) and two naphthoquinones (1,4-hydroquinone, 2-OH-1,4-hydroquinone).

The reaction was conducted at alkaline pH to reduce strength of Fenton's reagent (to prevent cleavage of benzene rings) and to improve solubility of humic materials, in particular, of coal humic acids in water medium. Three hydroquinones and two naphthoquinones (Figure 1d) were used in this work to modify leonardite humic acids (CHP) and peat fulvic acids (PFA). The polyphenolic compounds differed in redox potential: hydroquinones tend to have the higher $E_h$ values, whereas much lower values were characteristic for naphthoquinones.

The reaction was carried out in alkaline medium (pH 10–11), which enabled dissolution of HS and facilitated incorporation of quinones and hydroquinones into humic backbone. No visible change was observed in the reaction mixture during reaction. The obtained HA derivatives were black powders, while the derivatives of fulvic acids had a bright brown color. The obtained derivatives were characterized using [13]C-NMR and FTIR spectroscopy. The [13]C-NMR spectra are shown in Figure 2.

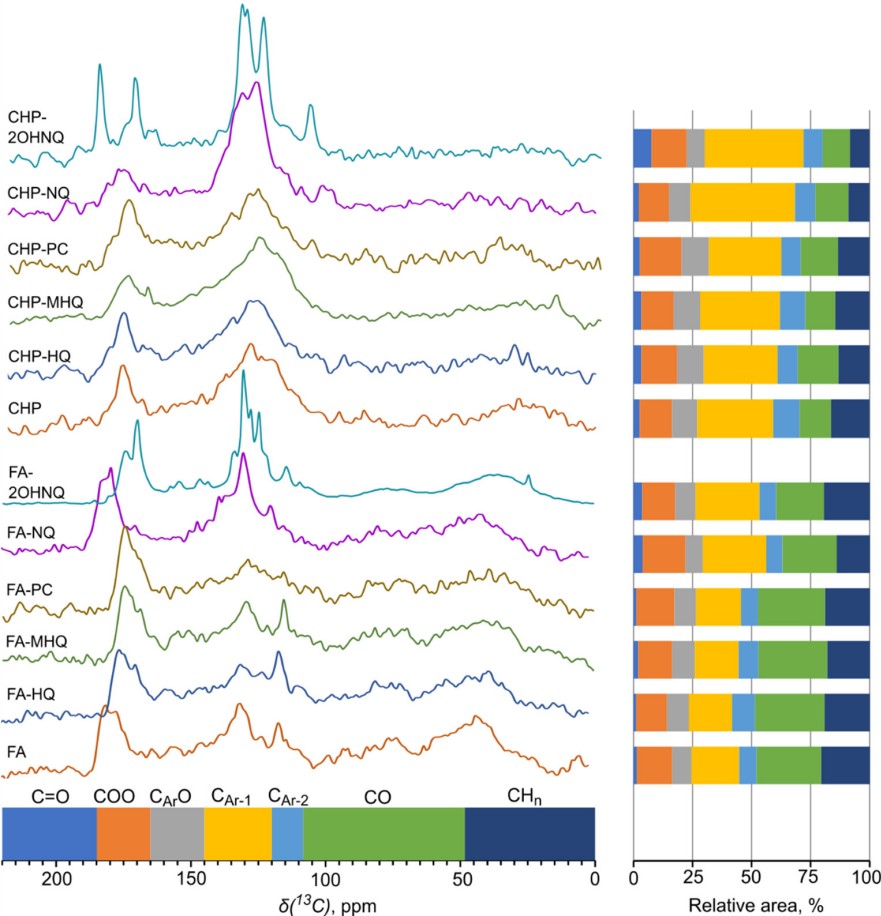

**Figure 2.** [13]C-NMR spectra of the parent humic acids (HA), fulvic acids (FA) and their derivatives with hydroquinones (1,4-hydroquinone, 2-methyl-1,4-hydroquinone, 1,2-hydroquinone) and naphthoquinones (1,4-hydroquinone, 2-OH-1,4-hydroquinone).

The shapes of the [13]C-NMR spectra were typical for coal HA [21,36–38]. They are characterized by high spectral intensity in the range of alkyl chains (0–45 ppm), aromatic structures (100–165 ppm), and carboxylic/ester carbon (165–185 ppm). Minimum intensity can be seen in the region of O-substituted aliphatic carbon (45–100 ppm). The CHP-NQ and FA-NQ derivatives were characterized with intense maximum in the region of 134 ppm characteristic of aromatic carbon atoms in the unsubstituted naphthoquinone ring. This is indicative of the presence of this structural group in the resulting derivative. For the spectra of hydroquinone derivative—CHP-HQ and FA-HQ there are change in the ratio of the intensities of the regions at 108–120 and 120–135, which can explain the occurrence of a fragment of hydroquinone in the modification, which has a signal at 115 ppm.

Typical FTIR spectra are shown in Figure 3. The spectra of both CHP and FA derivatives did not show the presence of sharp high-intensity bands characteristic of the low molecular weight modifiers—1,4-hydroquinone and 2-methyl-1,4-hydroquinone.

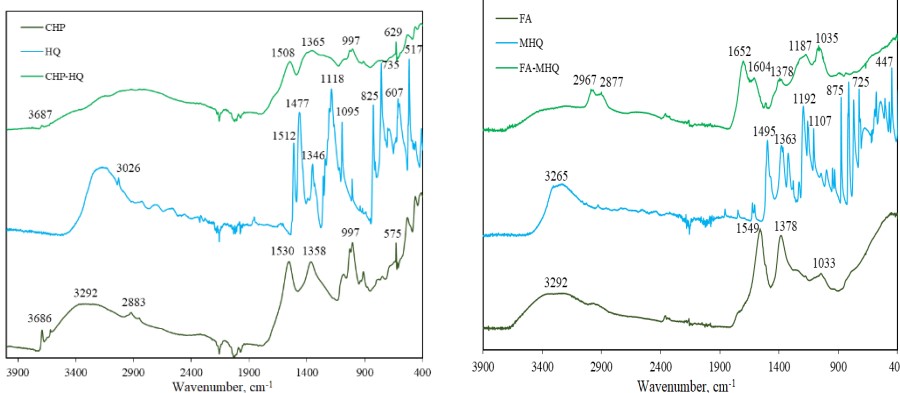

**Figure 3.** FTIR spectra of the parent humic acids (CHP), fulvic acids (FA), and derivatives with dihydroxybenzenes (1,4-hydroquinone and 2-methyl-1,4-hydroquinone).

The FTIR spectra of CHP and its derivatives show two broad absorption peaks at 1500–1550 cm$^{-1}$ and 1350–1400 cm$^{-1}$ characteristic of the carboxylate anion. An intense and broad peak in the region of 1000 cm$^{-1}$ can be attributed to silicate impurities of the parent potassium humate. The spectra of FA are characterized with intense absorption peaks at 2960, 2880 cm$^{-1}$, which can be attributed to the signals of stretching vibrations of the C-H methyl group in the modifier. The data of FTIR spectroscopy are indicative of the formation of modified derivatives of CHP and FA, while they cannot be considered as a mere superposition of the starting compounds.

Optical properties of the HS derivatives synthesized in this work were characterized using UV-vis and fluorescence spectroscopy. The spectra of HA and FA derivatives with naphthoquinones contain characteristic absorption bands of individual quinones and hydroquinones. The spectra of the CHP derivatives of 2-hydroxy-1,4-naphthoquinone contain a peak at 260 nm. At the same, the characteristic bands for derivatives with hydroquinones were not observed in the UV-Vis spectra. A comparison of the fluorescence spectra shows that the derivatives are characterized by the lower fluorescence intensity and wider range of emitted wavelengths compared to initial humic materials. Incorporation of additional hydroquinone centers which are not conjugated to the aromatic system of HA decreases conjugation of the derivatives and leads to an increase in the intensity and a blue shift of the fluorescence spectra to the region of shorter wavelengths (420–440 nm). At the same time, hydroquinone-substituted ones are characterized by a redshift of the spectrum (440–450 nm), associated with an increase in the aromaticity of the molecular ensemble of FC derivatives. Along with a change in the position of fluorescence, changes in the shape of the fluorescence band are of great diagnostic importance for characterizing the structural features of HA and FA. From the obtained fluorescence spectra, the following descriptors were calculated (Table 2).

It was found that the SUVA254 value, which characterizes the degree of aromaticity of HS, decreased in the order CHP > CHP-HQ > CHP-MHQ > CHP-PC for HA derivatives. The opposite situation was observed for FA derivatives modified with hydroquinones: the SUVA254 value increased in the order FA = FA-MHQ < FA-HQ < FA-PC. In accordance with a decrease in E2/E3 value, which is directly proportional to a size of the molecule, modification brings about an increase in the size of FA molecules. Fluorescence data demonstrate a significant difference in the shape of fluorescence spectra of HA and FA derivatives, in particular, in the red-wavelength range of the spectrum, so incorporation of additional hydroquinone- and naphthoquinone-moieties into molecular ensemble of FA and HA leads to an increase in the Asm350 (asymmetry index) value. This indicates a "blue" shift in intensity (a decrease in the red wavelength range) of the derivatives characteristic to low molecular weight hydroquinones and naphthoquinones.

**Table 2.** Optical descriptors obtained as a result of processing the UV-Vic and fluorescence spectra of the parent HA and FA and their derivatives.

| Sample | SUVA254, L/mgC $\times$ cm | E2/E3 | E4/E6 | $\Lambda$, nm | Asm350 | I350, L/mgC $\times$ cm | $\lambda$350, nm |
|---|---|---|---|---|---|---|---|
| CHP | 0.066 | 2.65 | 3.51 | 95 | 2.23 | $1.83 \times 10^7$ | 445 |
| CHP-HQ | 0.063 | 2.54 | 3.26 | 97 | 3.10 | $1.02 \times 10^7$ | 432 |
| CHP-MHQ | 0.052 | 2.67 | 3.30 | 101 | 3.20 | $1.44 \times 10^7$ | 440 |
| CHP-PC | 0.050 | 2.74 | 2.54 | 156 | 2.69 | $9.91 \times 10^6$ | 443 |
| CHP-NQ | 0.065 | 3.44 | 4.25 | 81 | 2.26 | $7.33 \times 10^6$ | 441 |
| CHP-2OHNQ | 0.059 | 3.77 | - | 83 | 3.47 | $1.96 \times 10^7$ | 422 |
| FA | 0.033 | 6.01 | 2.65 | 168 | 5.69 | $1.72 \times 10^7$ | 447 |
| FA-HQ | 0.038 | 3.45 | 2.66 | 113 | 6.13 | $1.69 \times 10^7$ | 431 |
| FA-MHQ | 0.035 | 4.61 | 3.90 | 74 | 5.86 | $8.81 \times 10^8$ | 442 |
| FA-PC | 0.038 | 4.66 | 3.50 | 94 | 5.66 | $1.09 \times 10^9$ | 442 |
| FA-NQ | 0.039 | 5.31 | 4.08 | 93 | 6.26 | $9.53 \times 10^8$ | 447 |
| FA-2OHNQ | 0.031 | 8.46 | | 36 | 6.23 | $8.15 \times 10^8$ | 442 |

The obtained data show that both NMR, FTIR, and optical spectroscopy characterize successfully general structural changes caused by the undertaken modification to molecular ensemble of HS. At the same time, they cannot discern structural changes within individual molecular components of HS due to complexity of humic molecular ensemble. For solving this problem, we have used high resolution mass spectrometry—Fourier Transform ion cyclotron resonance mass spectrometry (FT ICR MS) in combination with H/D labeling. The details on methodological approaches are reported in our previous publications [23,25,26].

*3.2. Characterization of Modification Products Using H/D Exchange Followed by FT ICR MS*

In search of more precise experimental tools which could provide experimental evidence that the proposed modification occurs in accordance with the proposed scheme (Figure 1), we have conducted modification of FA with a use of deuterated hydroquinone [²H]HQ. The labeling procedure is described in the Experimental section. This was done for enabling unambiguous identification of the presence of low molecular weight modifier (HQ) within the molecular components of the modified FA. In addition, deuteration was conducted for parent FA and hydroquinone yielding the samples of [²H]FA and [²H]HQ, respectively. We analyzed then molecular components of the deuterated products ([²H]FA and [²H]FA-HQ) for the presence of HQ-carrying components of the general formula $C_6O_2H_{4-x}{}^2H_x$ possessing at least one deuterium atom.

The molecular compositions of deuterated samples ([²H]FA-HQ, [²H]FA, [²H-HQ]) were determined using FT ICR MS and the assigned formulae were plotted into van Krevelen diagrams (Figure 4A,B). The D-labeled sample of FA-HQ ([²H]FA-HQ) was characterized with an expanded halo of the molecular components with the higher O/C and lower H/C values as compared to the parent FA ([²H]FA) (Figure 4B versus Figure 4A), which might be explained by progressive oxidation occurring during the condensation under the proposed conditions.

We have determined then the molecular components of the deuterated products ([²H]FA and [²H]FA-HQ) which carried $C_6O_2H_{4-x}{}^2H_x$ moiety with at least one deuterium atom and plotted in Figure 4C as color coded Van Krevelen diagrams. We found that the spectrum of the D-labeled HQ modified FA ([²H]FA-HQ) had 152 formulae of the parent FA modified with $C_6O_2{}^1H_2{}^2H_{2\text{-}}$. This is indicative of the covalent incorporation of the HQ fragments into FA components during the synthesis. Of importance is that the molecular components with the high O/C values were predominantly modified. These could be polyphenols-related structures, which form stable phenolate or phenoxyl radicals in the course of Fenton oxidation of phenols followed by polymerization reaction.

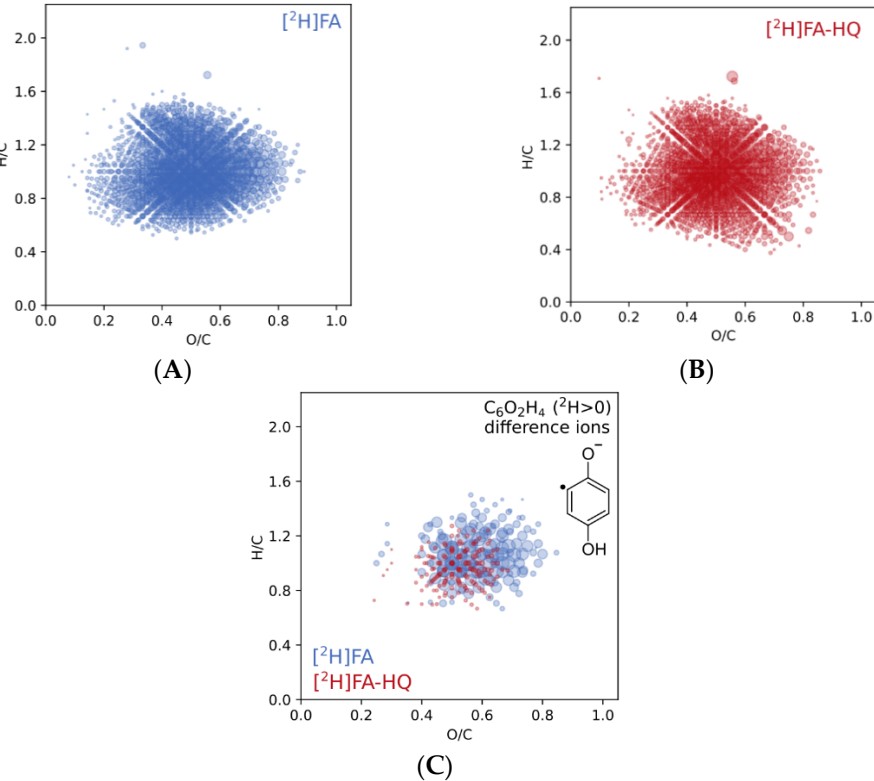

**Figure 4.** Van-Krevelen diagrams plotted from the FT ICR MS assignments for the deuterated samples (**A**) [²H]FA, (**B**) [²H]FA-HQ, and for their components modified by the fragment $C_6O_2H_4$ (²H > 0) (**C**). The diameter of the circles is proportional to the intensity of the peaks in the mass sheet.

### 3.3. Redox and Antioxidant Capacities of the Synthesized Humic Derivatives

Redox and antioxidant properties of the synthesized phenolic derivatives of HS were characterized using their ability to reduce $K_3[Fe(CN)_6]$ to $K_4[Fe(CN)_6]$ and to quench ABTS radical, respectively. The obtained values of reducing capacity (in mmol/g] and antioxidant capacity (AOE, Trolox equivalent per g] are shown in Figure 5A,B, respectively.

As it can be deduced from Figure 5A, all hydroquinone derivatives of HA and FA were characterized by a significant increase in the reducing capacity: from 0.2 to 1–2 mmol/g in the case of HA, and from 0.27 to 1–2 mmol/g, in the case of FA. The obtained values of redox capacity of the HA and FA and their derivatives are consistent with our previous studies [21] as well as with the values reported in the literature: The measured reducing capacity of CHP is 0.6 mmol/g which is corroborated by other data published for coal-derived humics [39,40]. These results are in the range of values reported under similar conditions for natural humics from soil (1.09 mmol/g), peat (2.29 mmol/g), and fresh water (6.5 mmol/g) [41].

At the same time, the naphthoquinone derivatives did not show significant changes in reducing capacity as compared to the parent humic materials except for CHP-2OHNQ whose reducing capacity dropped substantially. It is important to note that, in general, the redox capacity values of the parent FA and all FA derivatives were significantly higher as compared to those of the parent HA and all HA derivatives (except for catechol derivatives). This might be indicative of predominately donor properties of FA and acceptor properties of HA used in this study.

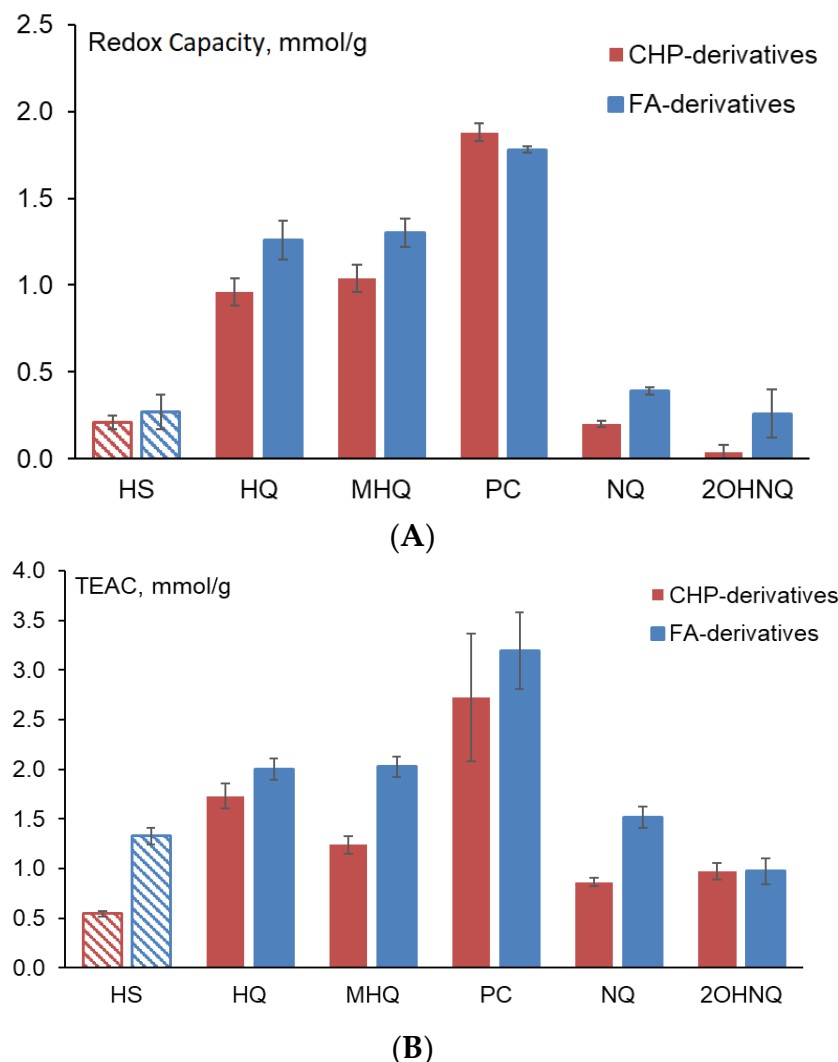

**Figure 5.** Redox capacity (**A**) and antioxidant capacity (**B**) of HA and FA derivatives with hydro-quinones (1,4-hydroquinone, 2-methyl-1,4-hydroquinone, 1,2-hydroquinone) and naphthoquinones (1,4-naphthoquinone, 2-OH-1,4-naphthoquinone).

Similar trends were observed for the AOE values measured with a use of the Trolox Equivalent Antioxidant Capacity (TEAC) method (Figure 5B). The antioxidant capacity of the FA derivatives outcompeted those of the HA derivatives. The obtained trends are in sync with those reported in [5] for electron donating capacities (EDCs), which accounted for 0.6 and 1.4 mmol(e)/g for Aldrich HA and SRFA, respectively—mmol(e)/g. They also corroborate well the findings on the antioxidant capacities of humic and fulvic acids determined with a use of ORAC method: the Aldrich HA was characterized with a value of 1.07 TEAC/g, whereas the riverine FA and HA (SRFA and SRHA) were characterized with a value of 1.22 and 2.07 mmol TEAC/g [42]. The similar range of values was also reported for different SPE isolates of marine DOM (0.5–2.0 mmol TEAC/g) [43].

As in case of reducing capacity, the maximum AOE values were characteristic for derivatives of the strongest donors—hydroquinone and catechol. At the same time, the AOE value of 2-methylhydroquinone derivatives of HA was significantly lower, and the lowest activity was observed for derivatives of 2-OH-1,4-naphthoquinone. A threefold increase in the AOE value of the catechol derivatives as compared to the hydroquinone derivatives could be explained by oxidative modification of these derivatives. The found values of the redox and antioxidant capacities were used for correlations analysis. The established correlation relationship is shown in Figure 6.

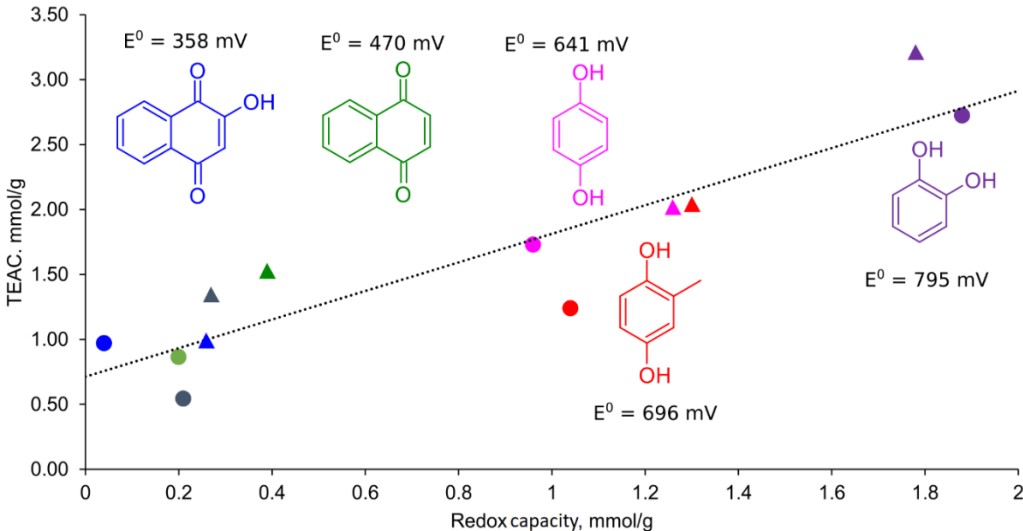

**Figure 6.** The relationship of antioxidant (TEAC) and reducing capacity of phenolic derivatives of HS. The colors denote: initial HA and FA (black), 2-OH-1,4-naphthoquinone (green), 1,4-naphthoquinone (blue), 2-methyl-1,4-hydroquinone (pink), 1,4-hydroquinone (red), 1,2-hydroquinone (purple) derivatives. Round marks—HA derivatives, trigonal marks—FA derivatives.

The obtained values of the reducing capacity and AOE yielded statistically significant linear correlation relationship (r = 0.913), which can be described by the following equation:

$$AOE\ (TEAC) = 1.1 + 0.7RC \tag{5}$$

where AOE (TEAC) is the antioxidant capacity determined by the TEAC method, mmol/g, RC is the reduction capacity (mmol/g), determined with respect to the Fe (II/III) redox pair in the hexacyanide complex.

The obtained relationship enabled a conclusion that the antioxidant capacity of the phenolic derivatives of HA and FA synthesized in this study was mainly determined by the reducing capacity of the phenolic modifiers. It is in sync with the findings of Aeschbacher et al. about the leading contribution of titratable phenols to the electron donating capacity of HS [5]. It means that incorporation of catechol either into HA or FA materials will bring about a maximum increase in the antioxidant capacity, it is followed by hydroquinone and 2-methylhydroquinone. In its turn, incorporation of 1,4-naphthoquinone and 2-hydroxy-1,4-naphthoquinone into either the HA or FA material will lead to a substantial drop into AOE due to acceptor properties of these compounds. Thus, we can conclude that the AOE capacity of the phenolic humic derivatives used in this study are determined by the nature of the phenolic redox center used to modify the HS matrix: the introduction of hydroquinones with high electrode potentials will lead to derivatives with enhanced reducing and antioxidant capacities, whereas incorporation of naphthoquinones with low values of electrode potential will reduce RC and AOE values of the parent humic material.

*3.4. Quenching Kinetics of ABTS by the Phenolic Derivatives of Humic and Fulvic Acids*

To determine amount of the fast and slow centers within the synthesized phenolic humic and fulvic derivatives, full kinetic curves were registered and fitted using Equation (3). The results are shown in Figure 6A,B. The corresponding values of the antioxidant capacity and the calculated amount of slow and fast centers in the phenolic derivatives of humic and fulvic acids (CHP and FA, respectively) are shown in Figure 7C–E.

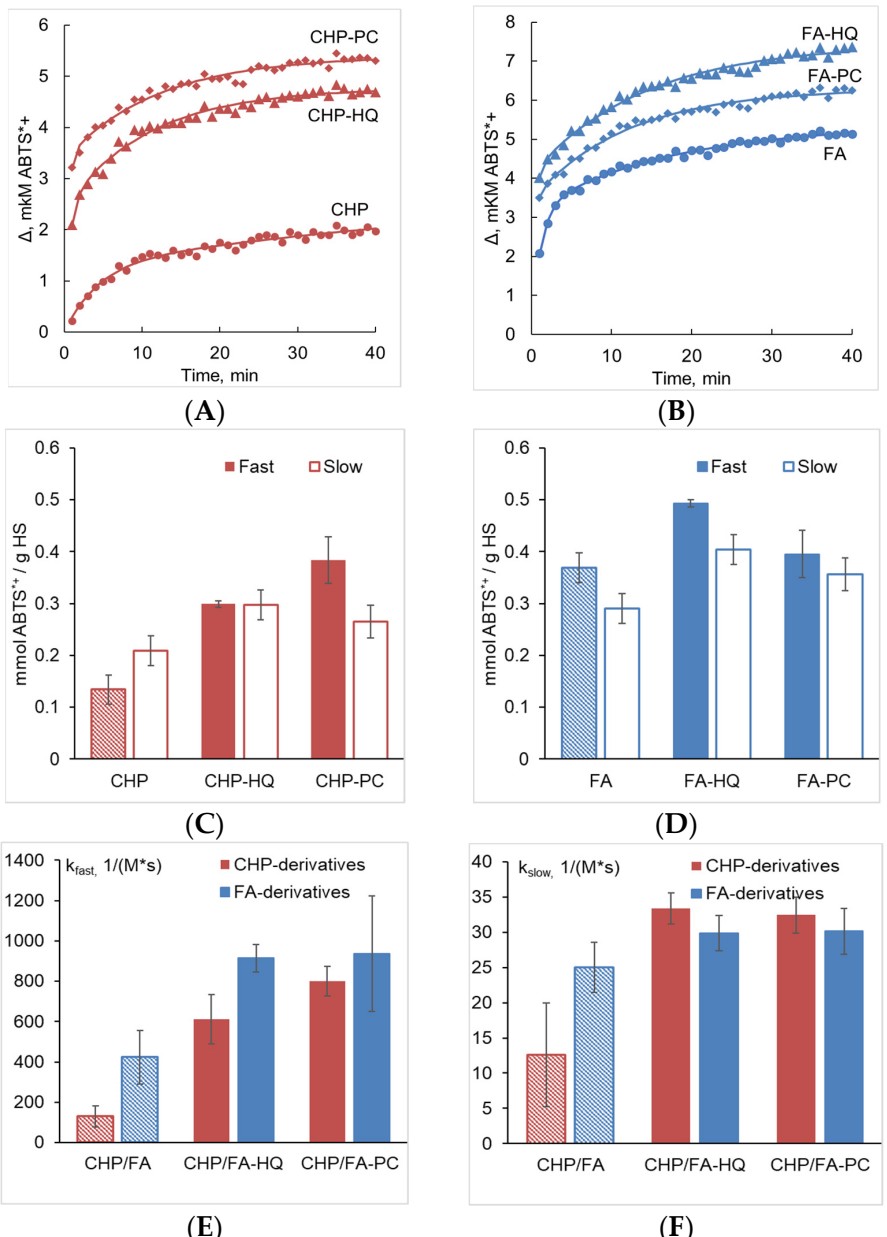

**Figure 7.** Kinetics of ABTS$^{\bullet+}$ quenching by phenolic derivatives of CHP (**A**) and FA (**B**) at a concentration of 8 mg/L. Points denote experimental dots, solid lines—fitting to Equation (2). The antioxidant capacity of the fast and slow parts of CHP derivatives (**C**) and FA derivatives (**D**), the rate constants of interaction between ABTS$^{\bullet+}$ and humic parts—the fast part (**E**), the slow part (**F**), (SD for N = 3).

As can be seen from Figure 7A,B, the model proposed by Klein et al. [33] yields satisfactory fits to the experimental data. The FA samples are much faster antioxidants as compared to the HA samples. When considering the parent HS, it can be seen that the coal HA quenched ABTS$^{\bullet+}$ radicals at the slower rate as compared to the peat FA [33–35]. This can be explained by the fact that HA, in general, and the leonardite HA, in particular, has a more condensed aromatic part containing sterically hindered phenols [36–38]. The sterically hindered phenolic antioxidants are characterized with the slower kinetics [31,43–45]. After modification with hydroquinones—HQ and CT, the quenching rate for all derivatives increased, the rate constants of both the slow and fast centers increased, indicating an increase in the contribution of the fast centers. This effect can be explained by an incorporation of additional fast centers via modification with HQ and CT, which are fast antioxidants.

## 4. Conclusions

The conducted study demonstrated the feasibility of the proposed approach to directly modify humic and fulvic acids by incorporation of hydroquinone and naphthoquinone centers via their in situ oxidation by Fenton reagent, followed by oxidative polymerization of phenols. The reaction mechanism was supported by D-labelling studies followed by FTICR MS analysis. Another major finding of this study is that the peat fulvic acid and its derivatives possessed much higher antioxidant capacity and the rate of quenching constants as compared to the leonardite humic acid. This can be explained by the natural differences of HA and FA: a much higher substitution degree and sterical hindrance of coal HA as compared to FA. In this case, the rate of antioxidant reactions will depend on the nature of the humic matrix: derivatives of HA humic acids will be characterized by a slowed reaction rate as compared to the derivatives of fulvic acids. At the same time, the type of modifier used for incorporation into the initial humic ensemble will define the value of its antioxidant capacity. It could be concluded that the directed modification of HS with phenolic centers is a promising tool for preparing humic antioxidants with the desired properties (Figure 8).

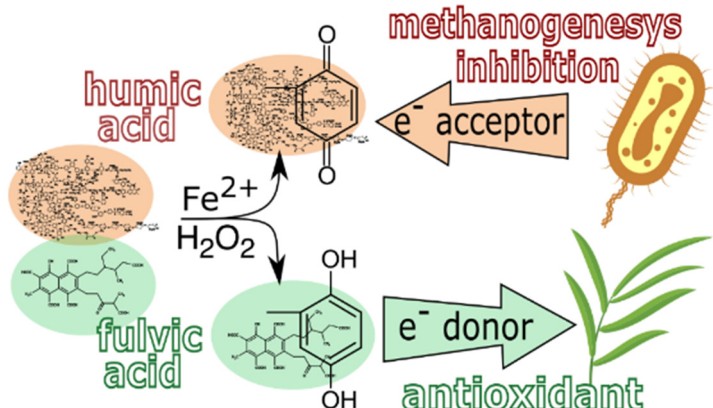

**Figure 8.** Humic and fulvic derivatives as electron acceptors and donors.

We have recently demonstrated good prospects of this approach for suppressing methane emission by consortium of methanogenic bacteria [46]: addition of 2-OH -naphthoquinone-HA-derivative into cultivation medium substantially reduced the rate and the amount of the emitted methane, while addition of fulvic acids and their hydroquinone derivatives stimulated the production of methane. From an agricultural perspective, it might be suggested that fulvic acids, in particular, those modified with hydroquinones, could possess enhanced electron-shuttling capacity, which is related to ability of humic substances for eliciting biostimulating activity on plant growth [47]. If slow-acting agents are needed, the preference should be given to humic acids and their derivatives.

**Author Contributions:** Conceptualization, I.V.P.; methodology, A.B.V., A.Y.Z., E.N.N. and D.S.V.; investigation, N.V.M., A.B.V., A.I.K., A.A.M. and S.V.M.; writing—original draft preparation, A.B.V., N.V.M., A.E.B. and M.E.G.; writing—review and editing, I.V.P., A.Y.Z. and M.V.Z.; visualization, A.B.V.; funding acquisition, I.V.P. All authors have read and agreed to the published version of the manuscript.

**Funding:** This work was funded by the Russian Science Foundation (grant #20-63-47070).

**Data Availability Statement:** Most data supporting the results are included in the article. The datasets used and/or analyzed during the current study are available from the corresponding author on reasonable request.

**Acknowledgments:** This work was funded by the Russian Science Foundation (grant #20-63-47070). Funding under state contract 121021000105-7 is appreciated. This research was conducted in the

framework of Interdisciplinary Scientific and Educational School of M. V. Lomonosov Moscow State University "Future Planet and Global Environmental Change".

**Conflicts of Interest:** The authors declare no conflict of interest.

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
