# Peer review of "Directed Synthesis of Humic and Fulvic Derivatives with Enhanced Antioxidant Properties"

_agronomy, doi:10.3390/agronomy11102047_

Round 1

Reviewer 1 Report

The article is of high quality: highly original concept, well elaborated experimental background, results obtained using up-to-date methods and their discussion. The study will have interest of readers. Some comments about the article:

  1. It would be good to use in figures the same writing of units as in text , for example, mg/L or mg L-1
  2. The conclusions are ending with the statement about possible biostimulant activity of obtained modified humics. However this statement is not based on evidences and thus I suggest to delete them.
  3. Writing of references in p 3
  4. Fenton reagent is quite strong and its action might result in destruction of original humic material. Such effects are not mentioned. 

Author Response

We thank the reviewer for thoughtful comments and enclose our answers to the reviewer's  criticism.

ANSWERS TO THE REVIEWER’S CRITICISM

Revision 1

The article is of high quality: highly original concept, well elaborated experimental background, results obtained using up-to-date methods and their discussion. The study will have interest of readers. Some comments about the article:

Q1. It would be good to use in figures the same writing of units as in text , for example, mg/L or mg L-1

A1. We have changed dimensions to mg/L throughout the text.

Q2. The conclusions are ending with the statement about possible biostimulant activity of obtained modified humics. However this statement is not based on evidences and thus I suggest to delete them.

A2. We have soften the statement and provided the recent review publication, which sets very high electron-shuttling  capacity for biological activity of humic substances on the plant growth (Larmar, 2020) [47]

Q3. Writing of references in p 3

A3. We have inserted brackets - thanks. 

Q4. Fenton reagent is quite strong and its action might result in destruction of original humic material. Such effects are not mentioned.

A4. We have added explanation to the manuscript:

Lines 72-75

«This study is devoted to development of an alternative “green” synthesis of the quinonoid-enriched derivatives, which uses Fenton’s reagent for in situ oxidation of hydroquinones to quinones under mild conditions followed by oxidative polymerization of the produced quinones with humic aromatic backbone.»

Lines 254-256

«The reaction was conducted at alkaline pH to reduce strength of Fenton’s reagent (to prevent cleavage of benzene rings) and to improve solubility of humic materials, in particular, of coal humic acids in water medium».

Reviewer 2 Report

Title: Directed Synthesis of Humic and Fulvic Derivatives with Enhanced Antioxidant Properties
Abstract: 
Line 14 -15: contains functional groups such as hydroxyl (phenolic and alcohol), carboxyl, carbonyl, quinone and methoxy groups
Although the abstract sought to capture the entire study, it is poorly written and difficult to understand. Please state clearly the aim or objective of the study and follow with the hypothesis, outcome, conclusion and recommendations if any in a free flowing concise passage. And check the grammar as well.
Introduction:
Line 33 – 35: The statement is unclear. Please rephrase and check the grammar.
The content of this section is fairly good but I find it poorly structured and the grammar should also be checked. 
I will suggest revising the entire section by properly incorporating previous similar studies before highlighting the main objective of this study.
Materials and Methods: 
Please check all equations. Make sure everything is written correctly e.g CHA?
I also suggest adding some photos of the experimental setups. 
Results & Discussion: I find the results part well presented. However, the whole discussion was mainly focused on the current study with almost no comparison with other similar studies.
Conclusion: This section needs some revision to make it concise. 
Although, the study is novel and interesting and will appeal to readers, I suggest the comments raised (and from all reviewers) are duly addressed to make it more comprehensible and concise to bring the quality to a publishable level.  I also suggest that the entire manuscript is checked with respect to the grammar.

Author Response

We thank the reviewer for thoughtful reading of the manuscript and constructive comments which helped us to improve quality of the manuscript.

Revision 2

Title: Directed Synthesis of Humic and Fulvic Derivatives with Enhanced Antioxidant Properties
Q1 Abstract: 
Line 14 -15: contains functional groups such as hydroxyl (phenolic and alcohol), carboxyl, carbonyl, quinone and methoxy groups
Although the abstract sought to capture the entire study, it is poorly written and difficult to understand. Please state clearly the aim or objective of the study and follow with the hypothesis, outcome, conclusion and recommendations if any in a free flowing concise passage. And check the grammar as well.

A1. We thank the reviewer. We have completely rewritten the abstract.

Its new version is given below:

“Redox moieties, which are present in the molecular backbone of humic substances (HS), govern their antioxidant properties. We hypothesized that a directed modification of humic backbone via incorporation of the redox moieties with the known redox properties might provide an efficient tool for tuning up antioxidant properties of HS.  In this work, hydroquinonoid and hydronaphthoquinonoid centres were used, which possess very different redox characteristics. They were incorporated into the structure of coal (leonardite) humic acids CHA) and peat fulvic acids (PFA). For this goal, an oxidative copolymerization of phenols was used. The latter was induced via oxidation of hydroquinones and hydroxynapjtaquinones with a use of Fenton’s reagent. The structure of the obtained products was characterized using NMR and FTIR spectroscopy. H/D labelling coupled to FT ICR mass spectrometry analysis was applied for identification of the reaction products as a tool for surmising on reaction mechanism. It was shown that covalent -C-C- bond were formed between the incorporated redox centers and aromatic core of HS. The parent humic acids and their naphthoquinonoid derivatives have demonstrated high accepting capacity. At the same time, the parent fulvic acids and all hydroquinonoid derivatives have possessed both high donor and antioxidant capacities. The kinetic study has demonstrated that both humic acids and their derivatives demonstrated much slower kinetics of antioxidant reactions as compared to fulvic  acids. The obtained results show, firstly, substantial difference in redox and antioxidant properties of the humic and fulvic acids, and, secondly, they can serve as an experimental evidence that directed chemical modification of humic substances can be used to tune and control antioxidant properties of natural HS.”

 Introduction:
Q2 Line 33 – 35: The statement is unclear. Please rephrase and check the grammar. The content of this section is fairly good but I find it poorly structured and the grammar should also be checked. 

I will suggest revising the entire section by properly incorporating previous similar studies before highlighting the main objective of this study.

A2. We have restructured the text mentioned by the reviewer.

Materials and Methods: 
Q3. Please check all equations. Make sure everything is written correctly e.g CHA?

A3. CHA is corrected. We have checked equations.

Q4. I also suggest adding some photos of the experimental setups.

A4. We do not have good photos of the experiments, unfortunately.

Q5. Results & Discussion: I find the results part well presented. However, the whole discussion was mainly focused on the current study with almost no comparison with other similar studies.

A5. We thank the reviewer for this comment. We have added the reported data and extended the corresponding discussion part as it is shown below:

Lines 404-409

The obtained values of redox capacity of the HA and FA and their derivatives are consistent with our previous studies [21] as well as with the values reported in the literature: The measured reducing capacity of CHP is 0.6 mmol/g which is corroborated by other data published for coal-derived humics [39, 40]. These results are in the range of values reported under similar conditions for natural humics from soil (1.09 mmol/g), peat (2.29 mmol/g), and fresh water (6.5 mmol/g) [41].

Lines 420-427

The obtained trends are in sync with those reported in [5] for electron donating capacities (EDCs), which accounted for 0.6 and 1.4 mmol(e)/g for Aldrich HA and SRFA, respectively - mmol(e)/g. They also corroborate well the findings on the antioxidant capacities of humic and fulvic acids determined with a use of ORAC method: the Al-drich HA was characterized with a value of 1.07 TEAC/g, whereas the riverine FA and HA (SRFA and SRHA) were characterized with a value of 1.22 and 2.07 mmol TEAC/g [42]. The similar range of values was also reported for different SPE isolates of marine DOM (0.5 – 2.0 mmol TEAC/g) [43].

Q6 Conclusion: This section needs some revision to make it concise. 

A6 We have revised conclusion section.

Q7 Although, the study is novel and interesting and will appeal to readers, I suggest the comments raised (and from all reviewers) are duly addressed to make it more comprehensible and concise to bring the quality to a publishable level.  I also suggest that the entire manuscript is checked with respect to the grammar.

A7 We tried to improve grammar and the layout of the manuscript.